

# A glimpse into the future of exposure and vulnerabilities in cities? Modelling of residential location choice of urban population with random forest.

Sebastian Scheuer[1], Dagmar Haase[1,3], Annegret Haase[2], Manuel Wolff[1], Thilo Wellmann[1,3]

[1]Landscape Ecology Lab, Geography Department, Humboldt-Universität zu Berlin, Berlin, 10099, Germany
[2] Department of Urban and Environmental Sociology, Helmholtz Centre for Environmental Research - UFZ, Leipzig, 04318, Germany
[3]Department of Computational Landscape Ecology, Helmholtz Centre for Environmental Research - UFZ, Permoserstr. 15, 04318 Leipzig, Germany

*Correspondence to*: Sebastian Scheuer (sebastian.scheuer@geo.hu-berlin.de)

**Abstract.** Disaster risk is conceived as the interaction of hazard, exposure, and vulnerability. Especially in urban environments, exposure and vulnerability are highly dynamic risk components, both being shaped by a complex and continuous reorganization and redistribution of assets within the urban space, including the residence of urban dwellers. This case study for the city of Leipzig, Germany, proposes an indirect, machine learning-based approach for the prediction of residential choice

behaviour to explore how exposure and vulnerabilities are shaped by the residential location choice process. The proposed approach reveals hot spots and cold spots of residential choice for distinct socioeconomic groups exhibiting heterogeneous preferences. We discuss the relationship between observed patterns and disaster risk through the lens of exposure and vulnerability, as well as links to urban planning. Avenues for future research include the operational strengthening of these linkages for more effective disaster risk management.

## 1 Introduction

In the human-environmental system, disaster risk arises from the interactions of different system components (Lian et al., 2015; Zscheischler et al., 2018). The Hyogo Framework for Action 2005-2015 maintains that disaster risk stems from the interaction of a hazard with exposed physical, socioeconomic and environmental vulnerabilities (UNISDR, 2007), consequently referring to the potential fatalities and losses in livelihoods, health, assets and services. Urban processes such as

the expansion into potentially hazardous areas, but also gentrification or densification shape exposure and vulnerabilities of services and assets within urban areas in a highly dynamic manner, and are thus at the basis of urban disaster risk. Hence, incorporating these urban processes more specifically into disaster risk assessment promises potential for more holistic perspectives.



Disaster risk $R$ is conceived as a function of the interacting, interdependent risk components hazard $H$, exposure $E$, and vulnerability $V$, expressed as $R = H$ x $E$ x $V$ (Brauch, 2011; UNISDR, 2015; Zscheischler et al., 2018). Here, hazard refers to potentially damaging physical events or latent conditions representing future threats of natural, human-natural (environmental) or human origin (UNISDR, 2007). Exposure denotes the physical aspects of disaster risk (UNISDR, 2004), referring to the socioeconomic and demographic spatiotemporal fabric, i.e., assets such as population or the built environment that is

potentially affected by a hazardous event (Brauch, 2011; Dilley et al., 2005; Villagrán de León, 2006). Vulnerability embraces the predisposition or propensity to be adversely affected, i.e., those physical, socioeconomic and environmental conditions leading to an (increase of) susceptibility or fragility of elements exposed to hazards (Carreño et al., 2017; UNISDR, 2007). Disaster risk is consequently driven by the specifics of hazardous conditions, i.e., hazard extent, severity and return period (Fuchs et al., 2013), as well as by (changes in) exposure and the degree of vulnerability (Cardona et al., 2012). In the case of

extreme events, disaster risk is mostly conditioned by exposure (UNISDR, 2015). $H$, $E$, and $V$ are dynamic over time and across spatial scales, and are thus non-stationary (Fuchs et al., 2013). This gives rise to considerable uncertainty in the assessment of future risks (Cardona et al., 2012; Sarhadi et al., 2016; Westra et al., 2010; Zscheischler et al., 2018), thus calling for a more holistic, combined assessment of all relevant risk drivers (Fuchs et al., 2013; Winsemius et al., 2016).

Whilst climatic drivers, encompassing both natural variability and anthropogenic climate change, affect the magnitude and (joint) probability of (compound) hazardous events (Carrão et al., 2018; Sarhadi et al., 2016, 2018; Zscheischler et al., 2018), non-climatic drivers including socioeconomic and demographic development with resulting land-use changes are shaping exposure as well as vulnerability (Elmer et al., 2012; Fuchs et al., 2013). Particularly high levels of or increases in exposure and vulnerability are found in the global urban land (Pelling, 2011; Scheuer et al., 2017). Urban areas as complex, highly

dynamic and integrated systems are particularly prone to hazards, which pose threats to physical assets as well as economic, social and political activities, disadvantaged populations and the urban poor, critical infrastructures, livelihoods and households (ibid.). This is due to various interlinking economic, social and spatial processes, e.g., the accumulation of capital, the increasing interconnectedness of places, increasing individualisation as well as urban growth and expansion (Cardona et al., 2012; Castells, 2002; Scheuer et al., 2017; UNISDR, 2015). For instance, from a global perspective, almost 90% of the

anticipated urban growth are expected in regions with limited economic development and thus comparatively high vulnerability including, e.g., the small-to medium-sized cities of Africa and Asia (Scheuer et al., 2016; Seto et al., 2013; UNDESA, 2019).

As recognized for instance by Castells (2002), Smith (2002) or Harvey (2009), these global phenomena are linked down to the

local level through their repercussions on the urban form. Consequently, also from this local perspective, exposure is governed firstly by urban population growth and the expansion of urban land. However, exposure is also shaped by multiple processes such as neighbourhood redevelopments and urban and economic restructuring, gentrification, infill and densification as well as urban decay, (intra-urban) mobility and (rural-urban) migration, as well as social-spatial segregation, increasing polarization





and growing inequalities (Braubach and Fairburn, 2010; Broitman and Koomen, 2015; Mustafa et al., 2018; Pelling, 2011;
Smith, 2002). In this context, urban disaster risk is also driven by demographic changes and shifts (ageing) as well as by the
impacts of conditions of the natural and built environment on human wellbeing and human health (Giles-Corti et al., 2016;
Hunter et al., 2019; Sarkar and Webster, 2017).

The aforementioned processes bring about substantial reorganization of urban structures and functions and the redistribution
of activities and assets in cities (Harvey, 2009). This also effects changes in individual self-selections, preferences, and
attitudes, e.g., regarding the choice of residential location and household mobility (Aslam et al., 2019). It has been estimated
that, overall, in North America, Australia and New Zealand, the share of households moving annually is about 15 to 20 per
cent, and 5 to 10 per cent in Europe (Knox and Pinch, 2010). Household mobility is typically characterized as a two-step
process, i.e., the decision to seek a new residence, and its actual selection (Kim et al., 2005; Knox and Pinch, 2010). A
comprehensive body of literature on residential choice adopts stated preference approaches and discrete choice modelling to
study this decision process and the corresponding determinants of residential location choice. This includes case studies, e.g.,
for Burkina Faso (Traoré, 2019), China (Wu, 2004), Colombia (Stokenberga, 2019), Germany (Heldt et al., 2016), Israel
(Frenkel et al., 2013), the Netherlands (Ettema and Nieuwenhuis, 2017), Pakistan (Aslam et al., 2019), or the UK (Kim et al.,
2005; Walker et al., 2002). McFadden (1978) describes the choice of housing location as a rational, complex decision based
on multiple dwelling characteristics such as the number of rooms or types of appliances, as well as location/neighbourhood
attributes such as proximity to green spaces or and the accessibility to places of work, commerce, education, and transportation.
It is consequently recognized that residential location choice, and hence residential mobility and migration, are driving (intra-
urban) spatial (re-)structuring, and thus exposure and vulnerabilities (Hunter, 2005; Kim et al., 2005; Wu, 2004).

Yin (2010) additionally highlights the role of land-use policies and population densities in the residential location choice
process, and the urban-rural gradient patterns emerging from this process.  A substantial body of research studies this nexus
of household perceptions on environmental amenities and disamenities—i.e., risks—and their role in residential location
choice (Braubach and Fairburn, 2010; Ewing et al., 2005; Hunter, 2005; cf. Zhang, 2010 for a comprehensive list of references).
For example, in case of less developed countries, in-migration and residential location choice within hazard-prone areas is
often the result of lacking coordination of urban development, informality of large parts of the residential sector, lack of
institutional capacities, failed risk governance, lack of financial capacities, housing market discriminations, lack of knowledge,
awareness and risk perception of disadvantaged populations (Hunter, 2005; Zhang, 2010). However, in case of the more
developed countries, it is also highlighted that risks and potential losses are often accepted due to locational benefits (ibid.) or
outweighed by environmental amenities such as riparian areas, lakeshores, or scenic views (Benson et al., 2000; Yin, 2010).

Most approaches that investigate the nexus between residential housing choice and hazard risk assume an indirect approach,
i.e., the hedonic price model and associated regression methods (Zhang, 2010). Hereby, physical housing attributes, locational



and neighbourhood characteristics as well as environmental attributes—such as the level of exposure, risk or expected losses—are considered in the derivation of a willingness to pay (Xiao, 2017). Whilst following Zhang (2010) some empirical findings

suggest that residents' willingness to pay is indeed lower in hazard-prone areas, it is also remarked that this evidence base is not at all clear-cut. Direct approaches, e.g., using household surveys, thus aim to directly identify the respondents' main motivations and decision factors for a specific location choice, and the role that hazard exposure and risk play into them (ibid.).

This paper seeks to bring together the study of residential housing choice and the school of natural hazard risk assessment by

another indirect, machine-learning based approach. Unlike the aforementioned approaches, it is not the focus of this case study to estimate the willingness to pay in the presence or absence of natural hazard risk. It is also not aiming to elicit risk awareness, e.g., of households on the move. Instead, it is proposed to explore the means and insights that residential location choice modelling offers for the identification of possible trends in or spatial hot spots of exposure and/or vulnerabilities, which is considered fundamental information for disaster risk assessment. The case study presented in this paper builds on a random

forest model previously implemented by Scheuer et al. (2018). In their case study, akin to a stated preference approach, Scheuer et al. (2018) modelled residential choice behaviour towards *hypothetical* apartment listings in the city of Leipzig, Germany, as the likelihood of a positive or negative decision outcome considering heterogeneity of preferences, i.e., the variation of housing preferences across individuals and socioeconomic groups (Hoshino, 2011). This case study goes beyond the previous work by making predictions of residential choice for *actual* real-estate data in the form of apartments advertised for rent on a common

internet platform called ImmobilienScout24 (Boelmann et al., 2019), and by spatializing these predictions to elicit spatial patterns of residential choice and their change over time. In so doing, this case study seeks to address the following research questions:

(i)     Does residential choice modelling allow to identify spatial patterns of exposure, e.g., hot spots of (vulnerable)

120          socioeconomic groups? How are these spatial patterns of exposure, and thus vulnerabilities, shaped by the

heterogeneity of preferences as a function of the socioeconomic status of urban dwellers?

(ii)    Can residential choice modelling contribute to the estimation of changes in exposure and vulnerabilities by

detecting trends in the spatial distribution of vulnerable groups?

In so doing, this case study aims to bring disaster risk assessment forward by making manifold and complex urban dynamics that shape the spatial distribution of urban dwellers and that consequently drive urban exposure and vulnerabilities more accessible in the assessment process.





## 2 Materials and Methods

Predictors for residential choice include spatial as well as non-spatial housing attributes, i.e., *inclusive rent*, *location*, *number of rooms*, *total size*, *furnishing characteristics*, *house type*—i.e., the structure type of the apartment building—and neighbourhood amenities such as the presence of major roads, urban green areas, or local suppliers. Additionally, various household attributes including income, employment status, qualification, and age, are used for this prediction. The spatialization of the random forest model by Scheuer et al. (2018) necessitates that the real-estate data provided by Boelmann

et al. (2019) is re-coded, e.g., regarding categorial predictor variables, and geolocated. Hence, the methodology applied in this case study embraces the following steps (cf. Fig. 1): (i) Extraction of non-spatial housing attributes, i.e., the characteristics of each actual apartment, from the scientific-use file provided by Boelmann et al. (2019; cf. Table 1); (ii) Determination of spatially homogeneous units for the geolocation of prediction targets; (iii) Determination of spatial housing attributes based on ancillary data (Table 1); (iv) Formulation of a set of socioeconomic profiles to account for heterogeneity of preferences

(Table 2); and (v) Application of the pre-trained random forest model to predict the likelihoods of positive residential choice outcomes. To evaluate changes in residential choice over time, this case study considers three time steps: 2008/2009, 2013/2014, and 2018/2019. In the following, each methodological step is described in more detail.

Figure 1


First, the non-spatial housing attributes *house type*, *number of rooms*, *furnishing features*, *inclusive rent* (rent including heating costs), *condition* and *total size* (Table 1) were determined from the apartment advertisements listed in the scientific-use file (Fig. 1(a)). As shown in Table 1, all housing attributes except *furnishing features* have a one-to-one cardinality, i.e., each advertised apartment has exactly one inclusive rent etc. A given apartment may however have multiple furnishing features, such as fitted kitchen, courtyard/garden and so forth. This constitutes a one-to-many relationship.


Second, prediction targets, i.e., the individual advertised apartments, need to be geolocated. The geolocation of each apartment typically corresponds to its address. However, in the provided scientific-use file, due to privacy protection, the actual address is anonymized and coded to a 1 km² grid cell location on the so-called INSPIRE grid. Such a coarse spatial resolution is rather

unsatisfying, particularly in complex urban environments. To overcome this limitation, we suggest increasing the spatial resolution through a mapping of apartment locations to so-called spatially homogeneous units (SHU). SHU were identified on the basis of a grid with a spatial resolution of 250 m x 250 m instead of 1000 m x 1000 m, i.e., each grid cell of the original 1 km² grid was divided into sixteen sub-cells. A SHU is characterized by the following properties: (i) residential land-use; (ii) a predominant (unique) house type; and (iii) the presence or absence of each individual spatial housing attribute. Areas of

residential land-use were determined from official topographic land-use data ATKIS (BKG, 2018). The predominant house type for each grid cell was subsequently derived by intersecting the 250 m x 250 m grid with a dataset by Haase and Nuissl





(2007), that describes the urban structure of the city of Leipzig by a combination of land-use and (residential) house types, e.g., "single and semi-detached houses" or "prefabricated housing estates". House types were consequently assigned to each grid cell of the 250 m x 250 m grid through the intersection. Then, the grid cells with common types of housing within each original 1 km² grid cell were merged, and in so doing, the SHU were identified (Fig. 2). As shown in Fig. 2, it needs to be noted that as the final delineation of each SHU depends on the predominant house type, the size of the resulting SHU must not correspond to a single 250 m x 250 m grid cell, but may comprise more than one sub-cell.

Figure 2

In a next step, each SHU was assigned spatial housing attributes, i.e., the presence or absence of *major roads* as well as of *neighbourhood amenities* green urban areas, pharmacies, and local suppliers. It follows that similar to *furnishing features*, *neighbourhood amenities* constitute an attribute with a one-to-many cardinality, where the presence of a given amenity was affirmed if at least 67 % of a SHU was within (cf. Fig. 1(b) and Table 1):

- a 150 m buffer area to major roads. This distance threshold is in line with literature that suggests that air pollutant concentrations are highest within this distance to major roads (Balmes et al., 2009) and is further supported by studies stating increased health risks—e.g., regarding obstetrical complications (Yorifuji et al., 2015), decreased lung function in adults (Balmes et al., 2009) or neurological diseases incidence (Chen et al., 2017)—within up to 200 m to major roads;

- the service area of urban green areas, defined by a walking distance of 250 m, a threshold in line with recommendations that urban green should be accessible within no more than a 300 m linear (buffer) distance or an approximately 5-minutes-walk (WHO Regional Office for Europe, 2016);

- the service areas of local suppliers or pharmacies, defined by a walking distance of 500 m or an approximately 10-minutes-walk (BBSR, 2015; Hoshino, 2011).

The advertised apartments were thence geolocated to a given SHU within their coded 1 km² grid cell by matching of house types.





Third, as a function of this geolocation, spatial housing attributes for each apartment listing were determined by the properties of the corresponding SHU. Moreover, the categorized *location* as well as *multiculturality* were determined (cf. Fig. 1(c) and Table 1).

Fourth, to account for heterogeneity of preferences, and in this way for different degrees of vulnerability (Table 2), predictions

are carried out for a set of socioeconomic groups that are characterized by *employment status*, *qualification*, *net income*, and *age* (Fig. 1(d)). In so doing, the shaping of exposure and vulnerabilities—and subsequently disaster risk—through residential choice can be illuminated as a function of these household characteristics. The attributes for each socioeconomic group were chosen from the factor distributions, i.e., mode, of the sampled dataset used by Scheuer et al. (2018) for random forest training. The hazard-specific degree of vulnerability, as exemplarily postulated in Table 2, is a compound based on the age and income

characteristics of each socioeconomic group. Regarding flood hazards, the estimated degree of vulnerability follows empirical findings by Steinführer and Kuhlicke (2007), whereas for heat stress, vulnerability is based on Heaton et al. (2014). In both cases, older persons feature generally higher degrees of vulnerability. Likewise, more deprived or disadvantaged groups feature higher vulnerabilities compared to less disadvantaged ones.






**Table 1: Types, variables, description, cardinality, and source of data.**

| | Variable | Description | Comment | Cardinality[1] | Source |
|---|---|---|---|---|---|
| Non-spatial housing predictors | Size | Classified size | The total size (area in m²) of the apartment. | 1:1 | Boelmann et al., 2019 |
| | Rooms | Classified number of rooms | The total number of rooms of each apartment. | 1:1 | |
| | Rent | Classified inclusive rent | The inclusive rent is the exclusive rent including heating costs in EUR. | 1:1 | |
| | Furnishing features | Availability of a courtyard/ garden, fitted kitchen, or insulation | Furnishing characteristics of the apartment. | 1:n | |
| | Condition | Classified condition of the apartment | Indicates if apartment is fully renovated (first occupancy [after reconstruction], like new, reconstructed, modernised, completely renovated), partly renovated (well-kept), or not renovated (needs renovation, by arrangement, dilapidated) | 1:1 | |
| | House type | Building structure type | Re-classified to factors Wilhelminian, Detached, GDR, and Post-reunification as required by the random forest model. | 1:1 | Haase and Nuissl, 2007 |
| Spatial housing predictor | Categorized location | Classified city district | Assignment of class as a direct function of the geolocation | 1:1 | Stadt Leipzig, Amt für Statistik und Wahlen, 2017 |
| | Neighbour-hood amenities | Major road (roads equivalent to types motorway, primary road, secondary road, tertiary road, trunk (including corresponding links) | 150 m buffer of major road | 1:n | OpenStreetMap, Staatsbetrieb Geobasisinformation und Vermessung Sachsen, Referat Geodatenservice |
| | | Urban green area (recreational and sport areas, shrubs, forests) | 250 m walking distance service area | | |
| | | Local suppliers | 500 m walking distance service area | | Self-compiled database |
| | | Pharmacies | | | |
| | | Multiculturality | Multicultural image as a function of categorized location | | Own classification |

[1] 1:1 indicates a one-to-one cardinality, 1:n indicates a one-to-many cardinality.



**Table 2: Set of household predictors in the form of socioeconomic profiles to represent societal groups differentially vulnerable and/or at risk.**

| Profile name | Socioeconomic characteristics | | | | Degree of vulnerability | |
|---|---|---|---|---|---|---|
| | Employment status | Qualification | Categorized net income (EUR) | Categorized age | Flooding | Heat stress |
| Young adults in education | In education | In education | < 600 | 20-30 | Comparatively lower | Comparatively lower |
| Academic professionals | Full-time employment | University degree | > 3100 | 30-40 | Comparatively lower | Comparatively lower |
| Middle-aged workers | Part-time employment | Skilled worker | 1100-1600 | 40-50 | Comparatively lower | Comparatively higher |
| Precarious unemployed persons | Unemployed | Without professional qualification | < 600 | 30-40 | Comparatively higher | Comparatively higher |
| Pensioner | Retired | Skilled worker | 600-1100 | 60-70 | Comparatively higher | Comparatively higher |

Fifth, applying the random forest model (Fig. 1(e)), the predicted probability $p$ for a positive residential choice is then a function of factor combinations: $p = f(house\ type,\ rooms,\ size,\ rent,\ features,\ location,\ amenities,\ employment, qualification,\ income,\ age)$. In this context, importantly, the pre-trained random forest model allows for only a single factor value per predictor variable. To overcome this limitation, for each apartment, the factor values of all predictors with a one-to-many cardinality—i.e., furnishing features $m$ and neighbourhood amenities $a$—were permuted to obtain all $a \cdot m$ factor combinations. E.g., a given apartment features both a garden and a fitted kitchen, so that $m = 2$. If this apartment is then located near both an urban green area and local suppliers, also $a = 2$, and predictions thus need to be carried out for all four possible combinations of factors, with the values of all remaining predictors being held constant. The predicted likelihoods of residential choice for all factor combinations were subsequently averaged per apartment, and thence aggregated at the level of SHU for further analysis, including hot spot and cold spot analysis using local G* statistics (Ord and Getis, 1995) as implemented in the R package *spdep* (Bivand and Wong, 2018).




## 3 Results

Figure 3 summarizes the non-spatial housing attributes of the advertised apartments. A total of N=25579 apartment listings were considered in this analysis; for the period 2008/2009, $n_{2008}$=5468, for 2013/2014 $n_{2013}$=10803, and for 2018/2019, $n_{2018}$=9308 (Fig. 4). In this context, it is important to note that this does not necessarily correspond to the number of apartments available for rent. Instead, a single apartment could be advertised multiple times, e.g., in case of short rental periods. The listings were geolocated to 132 different SHU, out of a total of 455 SHU identified across the whole city of Leipzig.

As shown in Fig. 3, listings include mostly apartments with a size between 40 m² and 80 m² and with two to four rooms. The highest share is of Wilhelminian house type—i.e., multi-storey tenement blocks—followed by buildings constructed in the GDR, i.e., prefabricated housing estates, and residential parks constructed post-reunification in the 1990s. In 2008 and 2013, a considerable number of apartments in GDR type housing was offered in rather bad condition, i.e., not renovated or requiring renovation. This number declined substantially until 2018. The majority of Wilhelminian housing is offered in good condition (fully renovated), although a considerable number is also categorized as only partially renovated. This is due to the rental object being categorized as only well-kept. Spatial housing attributes in form of the derived SHU properties including the *categorized location* and *multiculturality* as well as proximity to/presence of *neighbourhood amenities* major roads, urban green areas, local suppliers, and pharmacies, are visualized in Fig. 4.

Only 7 % of all SHU feature a multicultural image. Most SHU are attributed to be dominated by single or semi-detached housing (41.3 %), followed by multi-storey tenement blocks/Wilhelminian house types (33.4 %), prefabricated GDR housing estates (20.4 %), and post-reunification residential parks (4.9 %). This contrasts with the house types offered, that were majorly Wilhelminian style, whilst there are only few offers in single or semi-detached housing; this could also explain the high number of SHU in which no advertisements were geolocated. The median relative SHU area covered by the buffer area to major roads is equal to 60.8 % (mean 56.3 %); about 42.4 % of all SHU are considered to being within 150m to major roads. The median coverage of SHU by the service areas of urban green spaces is equal to 75.3 % (mean 67.1 %); more than half of the derived SHU (about 60.9 %) are located within 250 m walking distance to urban green. The median areas covered by the service areas of local suppliers and pharmacies are 3.6 % (mean 23.8 %) and 4.2 % (mean 27.3 %), respectively, so that only 14.9 % of all SHU are located within 500 m walking distance to local suppliers; for pharmacies, this share is equal to 18 %. Looking at Fig. 4, it becomes apparent that the coverage of SHU by local suppliers and pharmacies is concentrated in the city centre.

The demanded inclusive rent, averaged across the whole city, was equal to 477 EUR in 2008, 524 EUR in 2013, and increased further to 642 EUR in 2018. As shown in the Fig. 4, inclusive rents increased particularly in the central parts of the city, and to a lesser extent in the eastern parts of Leipzig. However, it is here where also a comparatively high number of apartments





are offered for rent. In the outskirts, particularly in the western parts of the city, inclusive rents remain lower, but so does the number of apartments listed for rent.

Figure 3

Figure 4

The predicted likelihoods for positive residential choices, averaged at the level of SHU per socioeconomic group as described in Table 2, were subsequently summarized to hot spots and cold spots using local G* statistics (Ord and Getis, 1995). Figure
5 shows the associated z-scores for the three considered time steps. Here, high z-scores (z > 1.65) indicate likely hot spots, i.e., a clustering of comparatively high likelihoods of positive residential choices for a specific socioeconomic group at a given location. Hot spots are therefore considered as featuring a comparatively high chance that a socioeconomic group moves into (or resides at) the location in question. Conversely, low z-scores (z < -1.65) indicate likely cold spots, i.e., a grouping of comparatively low likelihoods of positive residential choices. Consequently, cold spots are considered to feature lower chances
of a given socioeconomic group to move in.

In 2008, the spatial distribution of hot spots and cold spots between the different socioeconomic groups appears to be rather similar. In all cases, the western outskirts of the city comprising the district of Grünau, a prefabricated housing estate district with a rather negative image (Grossmann et al., 2015), is mostly avoided by all groups. Similarly, locations in the northern
outskirts feature relatively low z-scores across all socioeconomic profiles. However, until 2018, these patterns change considerably, thereby becoming less similar overall, with many of the changes being explained by "extreme" locations, such as the very city centre or the outskirts. The centre loses attractiveness, as indicated by decreasing z-scores. This is particularly true for middle-aged learned workers, precarious, unemployed persons, and pensioners, i.e., for the socioeconomic groups considered most vulnerable due to comparatively lower incomes, half-time-, precarious or lack of employment, and/or age. A
contrary trend of increasing z-scores suggests an increasing attractiveness of the corresponding locations. For these vulnerable socioeconomic groups, such a trend can be identified for previous cold spots such as Grünau in the west or locations in the north of the city.

In contrast to the more vulnerable socioeconomic groups, the spatial patterns of z-scores indicating hot spots and cold spots of
full-time employed academics and young adults in education appear to shift less over time. For these groups, the loss of attractiveness of the city centre is much less pronounced. It can be noted instead that certain hot spots, e.g., in the eastern parts of the city, seem to reinforce themselves. For these groups, it also appears to be the case that certain locations, e.g., Grünau in the western part of the city, remain rather unattractive, as indicated by continuously low z-scores over time (Fig. 5).

Figure 5





## 4 Discussion

This case study demonstrates that residential choice behaviour can inform disaster risk assessment through several means. First, it has been shown that the proposed methodology allows identifying hot spots and cold spots of residential choice for distinct socioeconomic groups, i.e., groups of population with heterogeneous preferences. Especially the hot spots of residential

choice are considered to highlight where a progressing concentration of the respective group of population is likely. Consequently, the spatial pattern of hot spots is seen to directly reveal the shaping of exposure and vulnerabilities towards specific hazards through residential choice processes. The impact on disaster risk becomes specifically obvious when the elicited hot spot/cold spot pattern is overlaid with hazard-prone areas to account for the hazard dimension of disaster risk. By so doing, areas of importance for disaster risk assessment can immediately be revealed. For example, Fig. 3 includes the area

potentially affected by a 1-in-300 years flood event, denoted as HQ300. By comparing this area with the pattern of hot spots, it appears that especially academics and young adults in education may be particularly exposed to flooding, a trend possibly explained by previous studies indicating that environmental amenities outweigh possible risks (Benson et al., 2000; Yin, 2010). Contrary to that, exposure and thus vulnerabilities to heat stress may be more dominated by the spatial patterns of the hot spots of the elderly and deprived socioeconomic groups (Heaton et al., 2014).


Second, it has been shown that the proposed methodology allows detecting changing patterns of residential choice behaviour over time, e.g., cold spots becoming more attractive, as well as hot spots "cooling", i.e., losing attractiveness. Particularly the former are considered to be of relevance in disaster risk assessment, as such "warming" cold spots could be highlighting spatial shifts of exposure and vulnerabilities, thereby possibly forming future hot spots of disaster risk. It is consequently such areas

that could pose a priority for intervention, and by bringing such potential hot spots to the attention of decision-makers, timely and proactive instead of rather reactive adaptation measures might be taken. In the case of heat stress, for instance, greening measures could be implemented for heat adaptation in evolving hot spots with low green space accessibility and thus lack of cooling potential (Andersson et al., 2020; Haase et al., 2019). Similarly, in the case of flooding, the implementation of both structural as well as non-structural (behavioural) flood protection measures may be facilitated. Such mitigation and adaptive

action address vulnerabilities and exposures (Cardona et al., 2012), thereby promising large potential for the reduction of damages and disaster risk (Winsemius et al., 2016).

Moreover, spatially co-located hot spots of residential choice for different disadvantaged socioeconomic groups may be highlighting a strong competition between these demand groups, and may furthermore be indicative of conflicts in urban

planning, e.g., due to diverging interests and needs between the development of residential areas for said demand groups vs. the implementation of greening as risk adaptation measure or for the improvement of environmental justice. It is consequently through such "feedbacks" that links between (the prediction of) residential choice behaviour, disaster risk assessment and urban planning become apparent, and the role of urban planning for managing disaster risks, climate change adaptation and



human health and well-being is emphasized clearly. In this context, the proposed method could point to relevant process-chains
between urban drivers, housing market dynamics, and disaster risk management, thereby inviting for research and action to
address developmental shortcoming or planning weaknesses.

Third, by providing disaster risk assessment with a spatially explicit model of residential choice, the spatial outcomes of a
multitude of urban processes influencing residential choice behaviour become incorporated into the disaster risk assessment
process. Thereby, additional bodies of knowledge are tapped into, and bridges built between different scientific disciplines. In
so doing, novel insights may be obtained allowing for a more holistic and integrative perspective on disaster risk, and a better
understanding of the importance of urban processes in driving and shaping of exposure, vulnerabilities and risks may be
achieved (Carreño et al., 2017). In the context of the presented case study, these processes include, e.g., (eco-)gentrification,
segregation, polarization, and aging, each influencing the formation of both hot spots and cold spots. In case of comparatively
privileged socioeconomic groups such as academics, hot spots may indicate an increasing (self-reinforcing) concentration of
potentially exposed (material, economic) assets at risk. For socioeconomically disadvantaged or more vulnerable groups of
people such as the unemployed or the elderly, hot spots may however put emphasis on locations of increasing socioeconomic
vulnerabilities. In contrast, cold spots reveal evasive behaviour of specific socioeconomic groups, e.g., due to increasing rents.
This becomes apparent in the wider city centre, which appears to be increasingly avoided over time by pensioners and the
unemployed who in turn shift, at least partially, towards the prefabricated GDR real-estate complexes such as Grünau (cf. Fig.
5). These findings are in line with previous case studies for Leipzig, e.g., on the centrally located Lene-Voigt-Park, where
greening lead to inner-city urban renewal resulting in an influx of higher-income family, rising rents and a subsequent exodus
of less privileged groups (Ali et al., 2020; Haase et al., 2017), or on the risk of accumulation of socially-weak and aging
population in the large prefabricated GDR housing estates (Brade et al., 2009). Hereby, the importance of selected predictors
in the shaping of patters of vulnerability and exposure is emphasized once more; for example, rent was identified to be amongst
the three most-important predictors of residential choice behaviour by Scheuer et al. (2018). Furthermore, it becomes clear
that the presented approach is a means for detecting and communicating social phenomena associated with complex urban
processes.

Whilst we believe that disaster risk assessment is brought forward by the proposed approach through informing the dimensions
of exposure and vulnerability by incorporating heterogeneous preferences of distinct sociodemographic and socioeconomic
groups, several shortcomings of the presented approach need to be identified. These include the overall data availability and
completeness of data, e.g., regarding neighbourhood amenities such as local suppliers, pharmacies etc. In this context, due to
re-using a pre-trained machine learning algorithm, also the choice of predictors and corresponding categorial values were
limited. Shortcomings further include the spatial resolution of the SHU for the geolocation of apartment listings, that is
obviously dependent on the way data was provided in the scientific-use file, but that is clearly too coarse to depict spatial
manifestations of 'hyper-local' urban processes such as redevelopments, retrofitting or urban infill (Xu et al., 2020) in high





detail, i.e., at site-level. The SHU's coarse spatial resolution thus compounds the quality of predictions of residential choice through the limited spatial representation of housing attributes, which had to be approximated at the level of SHU. For example,
in the case of house type, a dominant house type had to be elicited, thereby possibly neglecting other house types within a given grid cell.

In contrast to other case studies, also transferability is limited due to the reliance on case-study specific data, and due to local-specific patterns and trends being at play. However, the overall analytical lens of detecting patterns of residential choice based
on tacit knowledge, i.e., unconscious knowledge tied to personal experiences (Raymond et al., 2010) embedded into a broader setting of urban development is a unique approach which will be of increasing relevance for cities facing similar trends of built-up and climate changes (Scheuer et al., 2017). In this regard, and by revealing spatially explicit trends and shifts of heterogeneous groups of population and thereby enabling more precise ex-ante analysis, the proposed methodology could be particularly useful for urban planning authorities of cities in less developed countries, where census data is less reliably
available, thus calling for alternative data sources (Contreras et al., 2020).

It furthermore must be noted that the presented case study does not consider preferences or spatial attributes evolving over time, a limitation deriving from a lack of training data before 2018. Consequently, the residential choice predictions for the time steps 2008 and 2013 assume invariant (homogeneous) preferences, as well as a constant importance of predictors. This
shortcoming may however be alleviated by adapting the proposed methodology to enable continuous and incremental training—e.g., online random forests (Saffari et al., 2009) or Mondrian forests (Lakshminarayanan et al., 2014), each allowing for so-called online training—as part of long-term panel studies. Such longer-term studies could facilitate disaster risk assessment by further strengthening the linkages between urban planning and disaster risk management.

## 5 Conclusions

This paper proposes a methodology for the spatially explicit prediction of residential choice behaviour in the form of hot spots and cold spots for distinct socioeconomic groups, a process seen to (co-)govern spatial patterns of exposure and vulnerabilities, and subsequently disaster risk. Through the lens of predicting residential choice, the proposed methodology enables disaster risk assessment and management to improve (ex-ante) analysis of the highly dynamic spatial shifts and resulting distribution of urban population, and to tap into additional bodies of knowledge, e.g., through making heterogeneous preferences of
different socioeconomic groups accessible. In so doing, the assessment of exposure, vulnerabilities and disaster risk is brought forward. An interesting avenue for future research includes the revision of predictors alongside the perpetuation of the methodology to allow for online training. Thereby, additional components of vulnerability, exposure and disaster risk such as coping, preparedness or adaptation, could be incorporated more specifically. In so doing, linkages between the disaster risk



community and environmental justice, e.g., in the form of green space accessibility, shall be explored further and operationalized in more detail.

## Data availability

The scientific-use file with apartments listed for rent (Boelmann et al., 2019) can be obtained by delivery as indicated in the referenced DOI http://doi.org/10.7807/immo:red:wm:suf:v1.

## Author contributions

SS, DH, AH, MW and TW were responsible for conceptualization of the case study. Development and implementation of the methodology, formal analysis, and visualization by SS. Data acquisition by MW and SS. Writing – original draft by SS with contributions from all co-authors. Funding acquisition by DH.

## Competing interests

The authors declare that they have no conflict of interest.

## Acknowledgements

DH and MW were supported as part of the project ENABLE, funded through the 2015-2016 BiodivERsA COFUND call for research proposals, with the national funders The Swedish Research Council for Environment, Agricultural Sciences, and Spatial Planning, Swedish Environmental Protection Agency, German aeronautics and space research centre, National Science Centre (Poland), The Research Council of Norway and the Spanish Ministry of Economy and Competitiveness. In addition, DH and SS benefited from the GreenCityLabHue Project (FKZ 01LE1910A) and DH, SS and MW from the CLEARING HOUSE (Collaborative Learning in Research, Information-sharing and Governance on How Urban forest-based solutions support Sino-European urban futures) Horizon 2020 project (No 821242). SS  was additionally supported by the 2018 Summer Academy on World Risk and Adaptation Futures: Urbanization, hosted by The Institute for Environment and Human Security (UNU-EHS) of the United Nations University and the Munich Re Foundation (MRF). TW receives a scholarship by the Deutsche Bundesstiftung Umwelt DBU (German Federal Environmental Foundation).

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

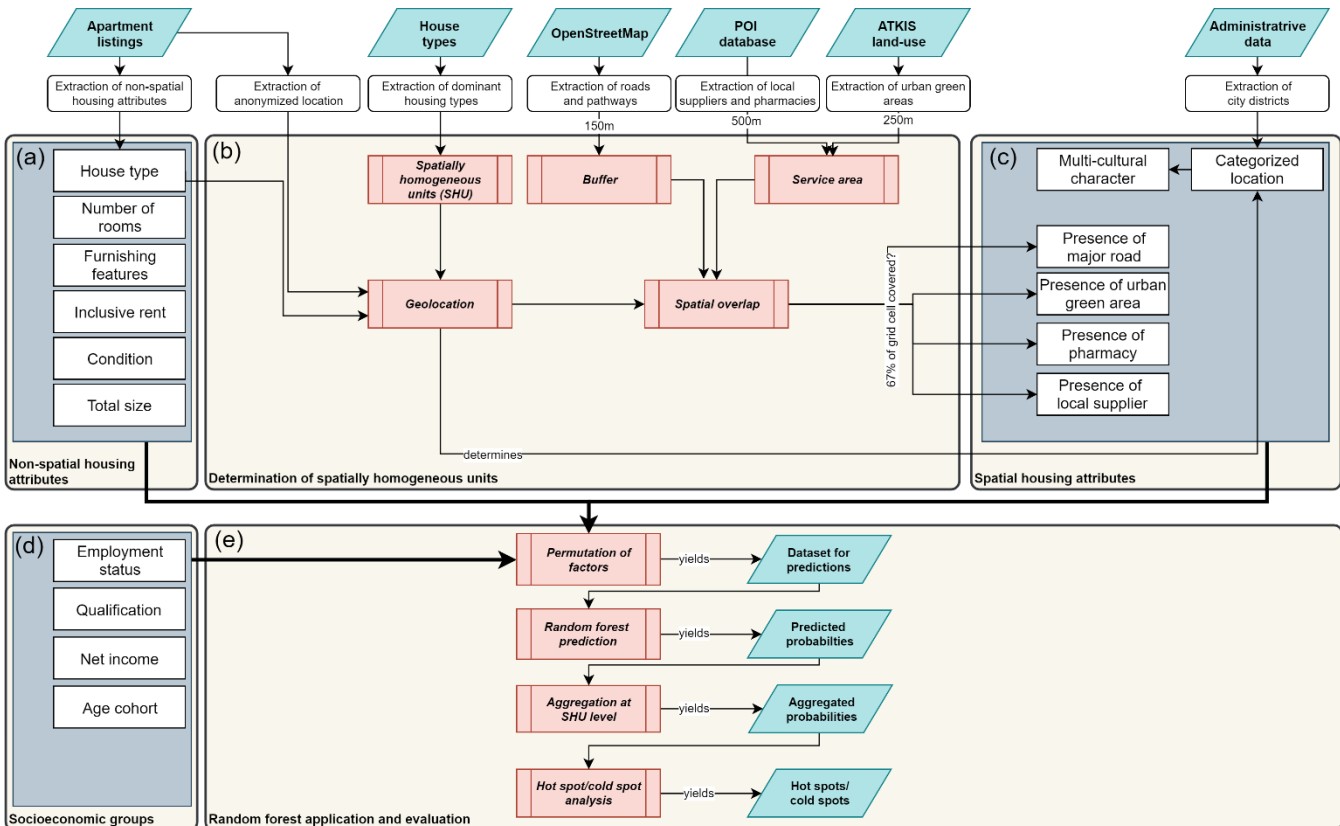

**Figure 1: Data (pre-)processing for the prediction of residential choice behaviour. (a) Non-spatial housing attributes are elicited directly from the apartment advertisements (Boelmann et al., 2019; cf. Table 1). (b) Identification of spatially homogeneous units, and estimation of neighbourhood amenities per spatially homogeneous unit based on the spatial overlap of buffer and service areas of major roads, urban green areas, pharmacies, and local suppliers. Geolocation of advertised apartments within these spatially homogeneous units; (c) Determination of spatial housing attributes as a function of the properties of the corresponding spatially homogeneous unit; (d) Based on a set of formulated socioeconomic profiles, household attributes are created. (e) Permutation of predictor factors and subsequent application and evaluation of the random forest model.**


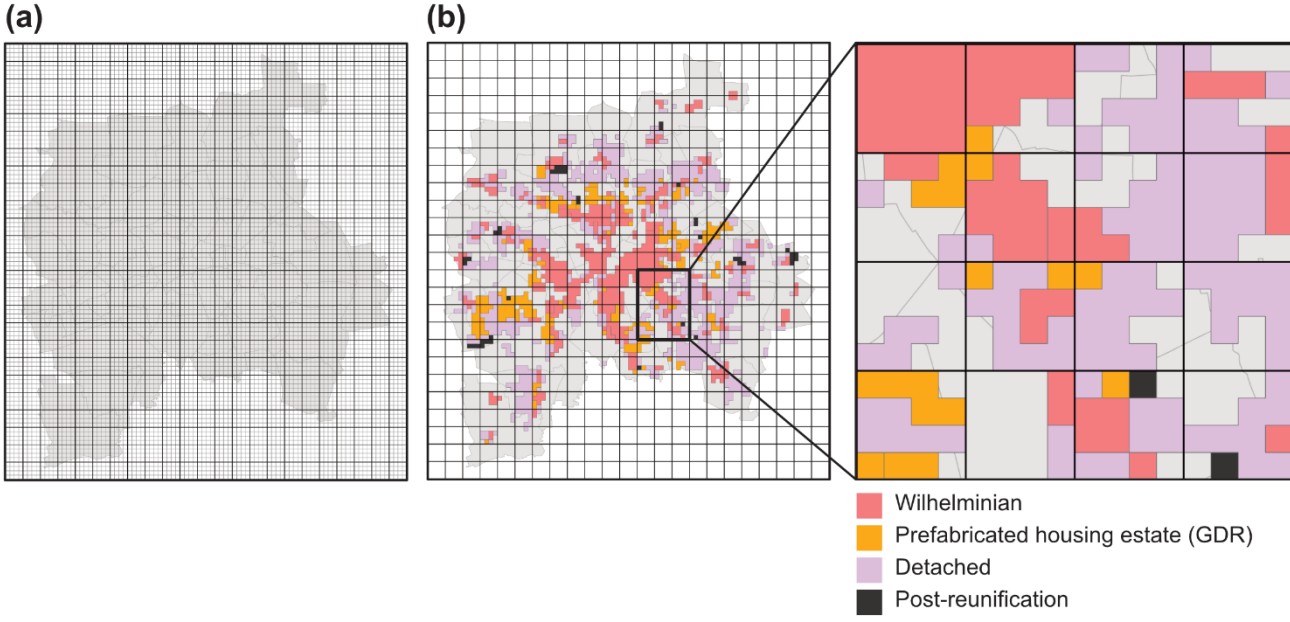


**Figure 2: Delineation of spatially homogeneous units (SHU) based on house types. (a) Study area overlaid with the 1 km² INSPIRE grid used in the scientific-use file (Boelmann et al., 2019) for geolocating advertised apartments (bold lines) and the 250 m x 250 m grid used as basis for the delineation of SHU (thin lines); (b) SHU obtained by dissolving the intersection of the 250 m x 250 m grid and the urban structure and land-use dataset by Haase and Nuissl (2007). As the detail shows, the final size of each SHU may vary**
**considerably, depending on the homogeneity/heterogeneity of urban structure and corresponding predominant house types within each INSPIRE 1 km² grid cell.**
**Figure 3: Characterization of the apartments offered for rent regarding predictors (a) categorized inclusive rent (rent including**

**heating costs); (b) house type; (c) categorized total size; and (d) categorized number of rooms. For house type, GDR is equal to prefabricated housing estates, Post-90 to buildings constructed post-reunification, and W to Wilhelminian-style buildings. For each house type, the condition is indicated in brackets: FR = Fully renovated; PR = Partly renovated; NR = Not renovated. Note that condition is not differentiated for post-reunification buildings due to the random forest training data.**

Natural Hazards
and Earth System

**Figure 4: Number of apartments offered for rent for each time step, identified SHU, spatial housing attributes per SHU, and inclusive rent (EUR) averaged per SHU per time step.**




**Figure 5: Map of local G\* z-scores indicating likely hot spots and cold spots of the predicted likelihoods of positive residential choices per socioeconomic group for the time steps 2008, 2013, and 2018. Arrows indicate at exemplary locations of persisting cold spots (a), reinforcing hot spots (b), hot spots turning into cold spots (c) and cold spots turning into hot spots for unemployed (d) or elderly persons (e). The map furthermore shows the area potentially affected by a 1-in-300 years flood event (HQ300).**
