# Peer review of "A glimpse into the future of exposure and vulnerabilities in cities? Modelling of residential location choice of urban population with random forest."

_Natural Hazards and Earth System Sciences, 2020_

## Referee Comment (RC1) · Philippe Ker Rault (Referee) · 30 Jul 2020

Excellent article, standard structure but very efficient. Well done.

---

## Short Comment (SC1) · 18 Aug 2020

Thank you very much for the kind feedback.
* * *

---

## Referee Comment (RC2) · Georgia Papacharalampous (Referee) · 31 Aug 2020

**Review Report**

| | |
|---|---|
| Journal: | Natural Hazards and Earth System Sciences |
| Reference code: | nhess-2020-213 |
| Title: | A glimpse into the future of exposure and vulnerabilities in cities? Modelling of residential location choice of urban population with random forest |
| Authors: | Sebastian Scheuer, Dagmar Haase, Annegret Haase, Manuel Wolff, Thilo Wellmann |
| Reviewer: | Georgia Papacharalampous |
| Date agreed to review: | 2020-08-15 |
| Date review submitted: | 2020-08-31 |
| Recommendation: | Minor revisions |

**Summary**

The paper proposes residential choice modelling for the indirect assessment of disaster exposure, vulnerability and risk. The investigations are carried out for the city of Leipzig, Germany by building on the work by Scheuer et al. (2018). In this respect, random forests by Breiman (2001) are used to predict the probability $p$ for a positive residential choice. The predictor variables include:

o   Spatial housing attributes, i.e., location and neighbourhood amenities.

o   Non-spatial housing attributes, i.e., size, rooms, rent, furnishing features and house type (including information on the apartment's condition).

o   Household attributes, i.e., employment status, qualification, income and age.

The random forest model has been pre-trained in Scheuer et al. (2018) by using interactive interview data from the same city; therefore, the real-estate data used in the present paper are accordingly re-coded and geo-located to support the prediction process. A switch from the INSPIRE grid (i.e., a grid with cells of dimensions 1 000 m x 1 000 m) to the spatially homogeneous units (SHU) grid (i.e., a grid with cells of dimensions 250 m x 250 m) is also made. In the SHU grid, each cell is characterized by the following properties: (i) residential land-use; (ii) a predominant house type; and (iii) the presence (or absence) of each individual spatial housing attribute. The formed dataset encompasses information for the years 2008/2009, 2013/2014 and 2018/2019, and for

five socio-economic profiles (i.e., young adults in education, academic professionals, middle-aged workers, precarious unemployed persons and pensioners). Each of the latter is known to be characterized by a specific degree of vulnerability, which might also be different for flooding and heat stress.

The predicted likelihoods are summarized in the form of hot and cold spots by using local G* statistics (Ord and Getis 1995) through the R package spdep (Bivand and Wong 2018). This is made separately for each set {year, socio-economic profile}. The resulted hot and cold spots are presented in maps, which allow the inspection of the changes observed as the years pass (separately for each socio-economic profile). Overall, it is demonstrated that residential choice modelling can be informative in disaster risk assessment and management.

**General comments**

In general, I find that the paper is meaningful, interesting, and very well-formulated and - written. I have only a few minor comments that could be addressed for improving the presentation of the already conducted work.

**Specific comments**

(1)   Since the paper uses random forests by Breiman (2001), this latter work should necessarily be cited, to my view.

(2)   Moreover, some basic information on random forests (see e.g., the review paper by Tyralis et al. 2019) should be provided (e.g., in an Appendix). This could be made by emphasizing the appealing properties of the utilized variants for the application of interest (see again the review paper by Tyralis et al. 2019). More generally, I feel that it would be particularly relevant to answer key questions like the following ones: Why are random forests selected in Scheuer et al. (2018) and herein? Could they be replaced by other machine learning algorithms?

(3)   It should also be noted that several references provided in Scheuer et al. (2018), such as Liaw and Wiener (2002), and Ishwaran et al. (2008, 2011), seem to be relevant in this paper as well. Currently, only the R package spdep is cited in the manuscript, while all the exploited R packages should be cited.

(4)   A short summary (additionally to lines 110–113) of the experiments carried out by Scheuer et al. (2018) could built some extra confidence in the use of the pre-trained random forest model. This summary could again be given in an Appendix.

(5) Furthermore, basic information on selected machine learning concepts could be provided. This information could be particularly important, given the technical character of the manuscript. The reader could also be referred to several specialized books (e.g., Hastie et al. 2009; James et al. 2013; Witten et al. 2007), for further information.

(6) The abstract could be revised to better reflect the novelty of the work. For instance, it could start with lines similar to the following: "The most common approach to assessing natural hazard risk is by investigating the willingness to pay in the presence or absence of such risk. In this work, we propose a new (also indirect) approach to the problem, i.e., through residential choice modelling".

(7) Some hints on how the title should be perceived could also be provided in both the abstract and the introductory section. For instance, one could think that the paper is about forecasting (which is not the case).

(8) Finally, there are very few typos in the manuscript. For instance, in Figure 2(b) the right big box (including 16 cells in the INSPIRE grid and 256 cells in the SHU grid) is larger by four cells in the INSPIRE grid than the one marked in the middle sub-figure of Figure 2. Another example exists in Table 2, in which "pensioner" should be replaced with "pensioners".

**References (not included in the manuscript)**

[1] Breiman L (2001) Random Forests. Machine Learning 45(1):5–32. https://doi.org/10.1023/A:1010933404324

[2] Hastie T, Tibshirani R, Friedman JH (2009) The elements of statistical learning: Data mining, inference and prediction, second edition. Springer, New York. https://doi.org/10.1007/978-0-387-84858-7

[3] Ishwaran H, Kogalur UB, Blackstone EH, Lauer MS (2008) Random survival forests. The Annals of Applied Statistics 2(3):841–860. https://doi.org/10.1214/08-AOAS169

[4] Ishwaran H, Kogalur UB, Chen X, Minn AJ (2011) Random survival forests for high-dimensional data. Statistical Analysis and Data Mining 4(1):115–132. https://doi.org/10.1002/sam.10103

[5] James G, Witten D, Hastie T, Tibshirani R (2013) An introduction to statistical learning. Springer, New York. https://doi.org/10.1007/978-1-4614-7138-7

[6] Liaw A, Wiener M (2002) Classification and regression by randomForest. R News 2(3):18–22

[7] Tyralis H, Papacharalampous G, Langousis A (2019) A brief review of random forests for water scientists and practitioners and their recent history in water resources. Water 11(5):910. https://doi.org/10.3390/w11050910

[8] Witten IH, Frank E, Hall MA, Pal CJ (2017) Data Mining: Practical machine learning tools and techniques, fourth edition. Elsevier Inc. ISBN:978-0-12-804291-5

---

## Short Comment (SC2) · 1 Sep 2020

Thank you very much for these helpful comments. We will address them and revise the manuscript accordingly as soon as possible. Kind regards!

---

## Author Comment (AC1) · 8 Oct 2020

Dear Editor, Dear Reviewers:

Please find described in this answer letter our replies and summarized changes made to the manuscript *nhess-2020-213*. In addition to the comments and suggestions raised by reviewers, we have carefully checked the manuscript again, and while doing so, moved all Figures from the end of the manuscript to within the body of text. We have also revised Figure 4 to improve readability. In the following, please find our detailed answers to the reviewers:

**Comments by RC1**

1. Comment: *Excellent article, standard structure but very efficient. Well done.*

   Reply: Thank you very much for this positive feedback.

**Comments by RC2**

1. Comment: *Since the paper uses random forests by Breiman (2001), this latter work should necessarily be cited, to my view.*

   Reply: This is indeed an unfortunate omission. We have included the reference to Breiman as suggested (cf. line 102).

2. Comment: *Moreover, some basic information on random forests (see e.g., the review paper by Tyralis et al. 2019) should be provided (e.g., in an Appendix). This could be made by emphasizing the appealing properties of the utilized variants for the application of interest (see again the review paper by Tyralis et al. 2019). More generally, I feel that it would be particularly relevant to answer key questions like the following ones: Why are random forests selected in Scheuer et al. (2018) and herein? Could they be replaced by other machine learning algorithms?*

   Reply: Thank you for this comment. We agree that it is very useful to include more detailed information on random forests in particular, and machine learning in general. However, we also feel that care is needed not to overstretch the scope of the manuscript. Trying to find a balance between these two points of view, we have included a statement on the advantageous properties of random forests as suggested (ll. 104–106), and also included additional references—the review of Tyralis et al. as well as Hastie et al.—for further reading (l. 106). We additionally emphasize that the random forest model used could be replaced by other methods of statistical learning (ll. 334–335), as has been pointed out. However, while doing so, we have refrained to go too in-depth to remain within the intended scope of the publication, which we see more on the application side of an existing model, and less so on discussing the optimal approach to creating said model. However, we emphasize a clear reference to the Scheuer et al. (2018) paper as the fundamental base of the current applied manuscript.

3. Comment: *It should also be noted that several references provided in Scheuer et al. (2018), such as Liaw and Wiener (2002), and Ishwaran et al. (2008, 2011), seem to be relevant in this paper as well. Currently, only the R package spdep is cited in the manuscript, while all the exploited R packages should be cited.*

   Reply: Thank you for this very valid point. The random forest that is re-use in the presented case study has been trained and evaluated using the *randomForestSRC* package described in *Ishwaran et al., 2008.* We have revised the manuscript so that the relevant R packages are included in the text, and we have added the citations accordingly (l. 184).

4. Comment: *A short summary (additionally to lines 110–113) of the experiments carried out by Scheuer et al. (2018) could built some extra confidence in the use of the pre-trained random forest model. This summary could again be given in an Appendix.*

   Reply: Fair point. Similar to your comment above, we agree that additional information may be beneficial. Following your suggestion, we have therefore included a statement on the overall performance of the model as published in Scheuer et al., 2018—cf. ll. 186–190. To keep the scope of the manuscript within feasible limits (also looking at cost of publication), we refrained from in-depth paraphrasing due to the level of detail and explanation required.

5. Comment: *Furthermore, basic information on selected machine learning concepts could be provided. This information could be particularly important, given the technical character of the manuscript. The reader could also be referred to several specialized books (e.g., Hastie et al. 2009; James et al. 2013; Witten et al. 2007), for further information.*

   Reply: As we will not be able to realistically provide a comprehensive review on machine learning methods within the scope of this manuscript, we followed your suggestion to refer the reader to the respective literature (ll. 106).

6. Comment: *The abstract could be revised to better reflect the novelty of the work. For instance, it could start with lines similar to the following: "The most common approach to assessing natural hazard risk is by investigating the willingness to pay in the presence or absence of such risk. In this work, we propose a new (also indirect) approach to the problem, i.e., through residential choice modelling".*

   Reply: Thank you for this suggestion. We have revised the abstract accordingly.

7. Comment: *Some hints on how the title should be perceived could also be provided in both the abstract and the introductory section. For instance, one could think that the paper is about forecasting (which is not the case).*

   Reply: We agree with your comment in that the case study presented it is not about a direct forecast per-se. However, we are convinced that the identification of previous and by extension ongoing trends may be a suitable proxy to forecast. The elicited trends reveal shifts of or reinforcements in spatial patterns that we consider a relevant contribution to explore the spatiotemporal dynamics of vulnerability and exposure. To emphasize our understanding, we have rephrased the abstract as suggested (l. 9) and, additionally, have emphasized this also in the introduction more clearly (l. 99).

8. Comment: *Finally, there are very few typos in the manuscript. For instance, in Figure 2(b) the right big box (including 16 cells in the INSPIRE grid and 256 cells in the SHU grid) is larger by four cells in the INSPIRE grid than the one marked in the middle sub-figure of Figure 2. Another example exists in Table 2, in which "pensioner" should be replaced with "pensioners".*

   Reply: Thank you for this comment. Unfortunately, these errors went unnoticed. We have changed the Figure in question accordingly and have also corrected the mentioned typos. The manuscript additionally underwent another round of careful proof-reading.

Sincerely,

Sebastian Scheuer, corresponding
author

---

## Author Comment (AC2) · 8 Oct 2020

Thank you very much again for your comments. We have addressed them in detail in the attached answer letter.

Please also note the supplement to this comment: https://nhess.copernicus.org/preprints/nhess-2020-213/nhess-2020-213-AC2-supplement.pdf

[Figure]

[Figure]

**(a)**  **(b)**

- Wilhelminian
- Prefabricated housing estate (GDR)
- Detached
- Post-reunification

**Fig. 1.** Revised Fig. 2

**(a) Trends in apartments offered for rent and demanded inclusive rents**

Listings count 2008

Inclusive rent (EUR) 2008

Listings count 2013

Inclusive rent (EUR) 2013

Listings count 2018

Inclusive rent (EUR) 2018

INSPIRE 1 km² grid    City of Leipzig

| | | |
|---|---|---|
| ■ 1 - 7 | ■ 23 - 67 | ■ 129 - 315 |
| ■ 8 - 22 | ■ 68 - 128 | |

| | |
|---|---|
| ■ 245 - 473 | ■ 561 - 637 | ■ 727 - 1350 |
| ■ 474 - 560 | ■ 638 - 726 | |

**(b) Spatially homogeneous units and spatialized predictors**

Prediction targets

Road network and SHU within 150 m to major road

House type

■ Spatially homogeneous units (SHU)

— Major roads
■ Area within 150 m to major roads

■ Wilhelminian   ■ Detached
■ GDR   ■ Post-Reunification

Urban green areas and SHU within 250 m walking distance to urban green

SHU within service areas (500 m walking distance) of local suppliers and pharmacies

Categorized location and multiculturality

■ Urban green area
■ Areas within 250 m walking distance

■ Service areas of local suppliers
■ Service areas of pharmacies

■ Centre   ■ Grünau   ■ West
■ East   ■ Outskirts
■ Gohlis   ■ South
■ with multicultural image

**Fig. 2.** Revised Fig. 4